# Guided Dental Implant Surgery: Systematic Review

**DOI:** 10.3390/jcm12041490

**Published:** 2023-02-13

**Authors:** Mario Dioguardi, Francesca Spirito, Cristian Quarta, Diego Sovereto, Elisabetta Basile, Andrea Ballini, Giorgia Apollonia Caloro, Giuseppe Troiano, Lorenzo Lo Muzio, Filiberto Mastrangelo

**Affiliations:** 1Department of Clinical and Experimental Medicine, University of Foggia, Via Rovelli 50, 71122 Foggia, Italy; 2Department of Precision Medicine, University of Campania “Luigi Vanvitelli”, 80138 Naples, Italy; 3Unità Operativa Nefrologia e Dialisi, Presidio Ospedaliero Scorrano, ASL (Azienda Sanitaria Locale) Lecce, Via Giuseppina Delli Ponti, 73020 Scorrano, Italy

**Keywords:** computer-assisted, survival, implant, PRISMA, dental implantology

## Abstract

Among the common procedures in clinical practice in the field of dentistry is prosthetic rehabilitation through the insertion of dental implants. In order to obtain the best aesthetic and functional results, the oral surgeon who deals with implantology must be able to position the dental implants correctly; a crucial role is therefore played by the diagnostic and treatment planning phases, where attention must be paid to anatomical constraints and prosthetic constraints in the alveolar bone site. The parameters, such as bone quality, bone volume, and anatomical restrictions, can be processed and simulated using implant planning software. The simulation of the virtual positioning of the implant can lead to the construction of a three-dimensional model of the implant positioning guide, which can be used during the implant surgery. The aim of this systematic review is to evaluate survival rates, early and late failure rates, peri-implant bone remodeling, and possible implant-prosthetic complications related to implants placed using digitally designed surgical guides. This systematic review was written following the indications of PRISMA and envisaged the use of 3 databases: Scopus, PubMed, and Cochrane Library. Results: Only 9 of the 2001 records were included, including 2 retrospective studies and 7 prospective studies. Conclusion: On the basis of the studies selected in this review, it can be seen that the implant survival obtained with the use of guided implant surgery shows high percentages. Many recorded failures occurred early, due to a lack of osseointegration, and the variables that come into play in the survival of the implants are many.

## 1. Introduction

Nowadays, the placement of osseointegrated dental implants is a common procedure in clinical dental practice. Implant therapy is guided by the restorative needs of patients and the technical and functional needs of each case; at the same time, this therapy can be limited by anatomical constraints. Therefore, correct implant placement is critical if an acceptable aesthetic and functional appearance of the restoration is to be achieved; furthermore, the positioning of the implant must respect the various critical anatomical elements often present in the vicinity of the site. Consequently, during diagnosis and treatment planning, the surgeon must pay close attention to both restorative and anatomical restrictions when selecting an alveolar bone site of adequate quality, thereby ensuring adequate and safe implant placement.

Various 3D diagnostic techniques are available, such as computed tomography and cone-beam computed tomography. The data relating to bone volume, bone quality, or anatomical restrictions can be processed and evaluated in the virtual implant simulation software.

This information allows the operator an active view of the anatomical structures within the jaw and is related to the radiological template and the future restoration. The placement of the virtual implant can be planned based on restoration objectives and anatomical limitations, which ultimately leads to the production of a guide model that can be used during surgery.

In recent years, several reviews have evaluated the accuracy of flapless guided surgery; some authors [1] have concluded that there is still no evidence that computer-assisted surgery is superior to conventional surgery in terms of safety, outcome, and efficiency. Three literature reviews (Voulgarakis et al., 2014 [2]; Lin et al., 2014 [3]; Moraschini et al., 2015 [4]) reported that implant survival ranges from 89% to 100%, even if the follow-up time is relatively short (6–48 months).

Velasco-Ortega et al. report in a very recent study on guided implantology conducted on 22 geriatric patients with 198 implants positioned in the mandible that the cumulative survival rate for all implants was 97.5%. The number of implants lost was 5, and the post-operative complications involved 10 patients with peri-implantitis and 6 with mechanical prosthodontic complications [5]. Furthermore, these data are partially confirmed by a subsequent study conducted by the same research group on a smaller number of patients and the implants with prosthetic overdentures positioned, reporting a 100% survival rate after an average follow-up period of 44 months [6,7,8,9].

Based on the data in the literature, it can be seen that guided flapless surgery is comparable to freehand surgery in terms of survival, marginal bone remodeling, and peri-implant variables [10].

The aim of this work is to evaluate survival rates, early and late failure rates, peri-implant bone remodeling, and any implant-prosthetic complications related to implants inserted using digitally designed surgical guides in the scientific literature of the last 10 years in order to evaluate the possibilities of applying guided surgery in implantology clinical procedures; moreover, the prevalence of biological complications such as peri-implantitis will also be evaluated.

## 2. Materials and Methods

### 2.1. Protocol and Registration

A systematic review of the literature regarding guided surgery in the implantology field was conducted, with particular, attention focused on results such as implant survival, early and late failure rates, bone remodeling for implants, and any implant prosthetic complications. The systematic review protocol has been registered on the INPLASY platform with registration number INPLASY2022110098 and DOI number 10.37766/inplasy2022.11.0098. The review was conducted following the guidelines of the Cochrane handbook, and the writing of the manuscript followed the indications of the PRISMA [11,12,13].

### 2.2. Information Sources, Search

In the period between November 1, 2022, and November 7, 2022, a systematic search was carried out independently by 2 reviewers in the PubMed, Scopus, and Cochrane Library databases with the implementation of the Science Direct and Google Scholar databases for the analysis of gray literature with the purpose of minimizing publication bias.

The keywords used were the following: computer-assisted, early and late failure rates, and bone remodeling. dental implant, prosthetic complications.

In particular, the terms used in detail on the PubMed search engine are the following:

Search: computer-assisted AND (early-late failure rate OR bone remodeling OR dental implant OR prosthetic complications) Sort by: Most Recent: “Computer-Assisted” [All Fields] AND ((“early” [All Fields] AND “late” [All Fields] AND (“failure” [All Fields] OR “failures” [All Fields]) AND (“j rehabil assist technol eng” [Journal] OR “rate” [All Fields])) OR (“bone remodeling” [All Fields] OR “bone remodeling” [MeSH Terms] OR (“bone” [All Fields] AND “remodeling” [All Fields]) OR “bone remodeling” [All Fields]) OR (“dental implants” [MeSH Terms] OR (“dental” [All Fields] AND “implants” [All Fields]) OR “dental implants” [All Fields] OR (“dental” [All Fields] AND “implant” [All Fields]) OR “dental implant” [All Fields]) OR ((“prosthetic” [All Fields] OR “prosthetically” [All Fields] OR “prosthetics” [All Fields]) AND (“complicances” [All Fields] OR “complicate” [All Fields] OR “complicated” [All Fields] OR “complicates” [All Fields] OR “complicating” [All Fields] OR “complication” [All Fields] OR “complication s” [All Fields] OR “complications” [MeSH Subheading] OR “complications” [All Fields]))).

Translations: failure: “failure” [All Fields] OR “failures” [All Fields]. Rate: “J Rehabil Assist Technol Eng” [Journal:_jid101671667] OR “rate” [All Fields]. Bone remodeling: “bone remodeling” [All Fields] OR “bone remodeling” [MeSH Terms] OR (“bone” [All Fields] AND “remodeling” [All Fields]) OR “bone remodeling” [All Field]. Dental implant: “dental implants” [MeSH Terms] OR (“dental” [All Fields] AND “implants” [All Fields]) OR “dental implants” [All Fields] OR (“dental” [All Fields] AND “implant” [All Fields]) OR “dental implant” [All Field]. Prosthetic: “prosthetic” [All Fields] OR “prosthetically” [All Fields] OR “prosthetics” [All Fields]. Complications: “complicances” [All Fields] OR “complicate” [All Fields] OR “complicated” [All Fields] OR “complicates” [All Fields] OR “complicating” [All Fields] OR “complication” [All Fields] OR “complication’s” [All Fields] OR “complications” [Subheading] OR “complications” [All Fields].

Furthermore, a last update of the records and keywords used was carried out on the databases on 16 December 2022.

### 2.3. Selection of Studies

The selection of the studies was conducted independently by two authors, with the task of evaluating the situations of doubt and conflict between the two authors. The selection of the studies and their relative inclusion were based on the application of the inclusion and exclusion criteria.

In order to assess the suitability of the studies, all the titles and abstracts of the publications generated by the research were consulted. The full text of the articles was retrieved in studies that appeared to meet the screening criteria and in studies in which the title and abstract did not give sufficient information to firmly decide whether to include the study or not.

Inclusion criteria: All guided surgery studies in the implantology field were included, in which there were data regarding implant survival, marginal bone resorption, intraoperative complications, and postoperative complications; the selected studies included observational, prospective, retrospective, RCTs, and multicenter studies.

Exclusion criteria: animal studies, guided surgery studies involving other anatomical areas, systematic reviews and meta-analyses, case reports, in vitro studies, and studies without abstracts in English were excluded.

### 2.4. Data Extraction

In each study, the data relating to the first author, year of publication, and journal were extracted, as were the following: study design, number of patients, age of patients, number of implants, diameter and length of the implants, duration of follow-up, state of the dental arch to be rehabilitated (edentulous partial or total), conditions of the implant site (post-extraction or healed), type of surgery (flap/flapless/miniflap), loading protocol, loading time with the final restoration, number of failures, implant survival rate, mean marginal bone loss, and intra- and postoperative complications.

The extraction of these data shows the data relating to the survival rate of the implants in relation to the distribution of the variables considered.

### 2.5. Risk of Bias

The ROBINS-I tool was used to calculate the risk of bias, and it was evaluated by the 2 reviewers (M.D. and A.B.) appointed to select the studies. The studies with a high risk of bias were excluded from the review.

## 3. Results

The whole selection process for the articles is described in Figure 1. The articles included in the systematic review are nine:✓Two retrospective studies: Polizzi and Cantoni, 2015 [14]; Meloni et al., 2010 [15].✓Seven prospective studies: Ciabattoni et al., 2017 [16]; Derksen et al., 2019 [17]; Lopes et al., 2015 [7]; Marra et al., 2013 [18]; Pozzi et al., 2012 [19]; Vogl et al., 2015 [20]; Yamada, 2015 [21].

In Table 1, the data relating to the study design, the number of patients, implants, the diameter and length of the implants, and the duration of follow-up were collected.

The most recent study included in this review is that of Derksen et al. (2019) [17], and the least recent one is that of Meloni et al. (2010) [15].

The number of patients included among the various studies varies from a minimum of 15 (Meloni et al., 2010 [15]) to a maximum of 66 (Derksen et al., 2019 [17]); the average age in the various studies is comparable.

For the rehabilitation of the totally or partially edentulous arches, a variable number of implants was inserted, as shown in Table 1; Pozzi et al. (2012) [19] evaluated both tilted and not tilted implants, while Vogl et al. (2015) [20] evaluated implants with immediate occlusal loading (defined as full loading at maximum intercuspidation) compared to implants without occlusal loading; the diameter and length of the implants used were then reported; only in 3 studies was it not possible to trace the value (Ciabattoni et al., 2017 [16]; Polizzi and Cantoni, 2015 [14]; Vogl et al., 2015 [20]).

The follow up of the studies ranges from a minimum of 1 year to a maximum of 5 years.

Table 2 contains the data relating to the state of the dental arch, the conditions of the implant site, the type of surgery (flap/miniflap/flapless), the loading protocol, and the prosthetic time with the final restoration.

In almost all the studies analyzed, guided surgery was used for the rehabilitation of totally edentulous arches; the implants were inserted both in post-extraction sites and in healed sites.

Flapless surgery, characteristic of guided surgery, was used in all studies, although the use of open flap surgery was reported in one study (Derksen et al., 2019 [17]) and miniflaps in another study (Pozzi et al., 2012 [19]). Loading was immediate in all papers except Derksen et al., which used conventional loading.

The placement of the definitive restoration took place between 4 and 8 months; this data has not been reported in a study (Ciabattoni et al., 2017 [16]).

Table 3 shows the number of implant failures, survival rate, mean marginal bone loss, and implant-prosthetic complications.

Both early failures as early as 2 weeks after insertion (Yamada et al., 2015 [21]) and late failures after 3 years (Ciabattoni et al., 2017 [16]; Lopes et al., 2015 [22]) were found; in the work of Vogl et al., (2015) [20], no cases of implant failure were reported.

The survival rate in the various studies varies from a minimum of 96.3% after 3 years (Pozzi et al., 2012 [19]), where there was early failure in five cases, to a maximum value of 100% after a follow-up of 1 year (Vogl et al., 2015 [20]).

The mean marginal bone loss ranges from a minimum value of 0.32 mm after a 1-year follow-up (Yamada et al., 2015 [21]) to a maximum value of 1.9 mm (Lopes et al., 2015 [22]; Marra et al., 2013 [18]); in the study by Derksen et al. (2019) [17], this value was not reported.

The implant complications reported in the various studies are:✓Intraoperative (surgery) complications, such as the impossibility of using a drill due to limited opening of the mouth of a patient, buccal bone dehiscence after osteotomy in another patient (both cases in the study by Derksen et al., 2019 [17]), insufficient bone quantity in one patient, and insufficient primary stability in three patients (Vogl et al., 2015 [20]; these cases were not considered in the study statistics, thus resulting in a 100% survival rate).✓Postoperative, (prosthetic implant complications) include loss of implants in almost all studies, loosening of the abutment screw in 2 cases in the study by Lopes et al., 2015 [7], in 3 cases in the study by Vogl et al., 2015 [20], in 10 cases in the study by Yamada et al., 2015 [21]; fracture of the definitive prosthesis in 7 cases in the study by Lopes et al., 2015 [22], in 9 cases in the study by Marra et al., 2013 [18], in 2 cases in the study by Vogl et al., 2015 [20]; fracture of provisional prostheses in 2 cases in the study by Meloni et al., 2010 [15], in one case in the study by Yamada et al., 2015 [21]; imperfect fit of the provisional prosthesis in 2 cases in the study by Meloni et al., 2010 [15], in 3 cases in the study by Vogl et al., 2015 [20]; need for occlusal adjustments in 2 cases in the study by Vogl et al. 2015 [20].

On the basis of the data obtained from the studies examined, it was possible to calculate the value of the early and late failure rate of each study and that relating to the total of the studies analyzed, at the implant level (implant level) and at the patient level (patient level). The latter data was possible to obtain by excluding two studies (Ciabattoni et al., 2017 [16]; Marra et al., 2013 [18]). As in these it was not possible to trace the number of patients in which failure had occurred; the data relating to the implant level were obtained taking into account the number of implants that failed early (or late) out of the total number of implants inserted: (number of failed implants × 100)/total implants inserted; the data relating to the patient level were obtained considering the number of patients in which failure (early or late) occurred out of the total number of patients present: (number of patients with failure × 100)/total patients with implants.

In the study by Vogl et al. (2015) [20], no cases of implant failure were recorded. The data used for the calculation were the following:

Total number of implants inserted: 1495

✓Total early failures: 24✓Total late failures: 7✓Total patients: 226✓Patients with early failure: 9✓Patients with late failure: 2

Table 4 shows the early and late failure rates in each study and the overall early and late failure rate, at the implant level:✓Total early failure rate: 1.60%✓Total late failure rate: 0.47%

Table 5 shows the early and late failure rates for each study, and the overall early and late failure rate at the patient level:✓Total early failure rate: 3.98%✓Total late failure rate: 0.88%

The data on peri-implantitis and, more generally, on the biological complications are reported in detail in Table 6. The most commonly reported post-operative biological complication, which is also the final event that leads to the loss of the implant, is the lack of osseointegration. The presence of peri-implantitis is reported in at least 4 studies and appears to be the main complication. An additional consideration should be made on the consumption of smoked tobacco, it should be noted that the early loss of the implant due to lack of osseointegration occurred in smoking patients in at least 3 studies.

The data related to smoking are reported in Table 6, and I provide a brief summary below.

Ciabattoni reports heavy smokers with a daily consumption of more than 10 cigarettes as exclusion criteria. He also advises smokers not to smoke for at least a week after surgery [16].

Derkesen reports early mandibular implant failure in a smoker, and smoking was not an exclusion criterion [17].

Lopes included two smokers who had more marginal bone loss after 3 and 5 years of follow-up and further implant failure. In addition, there were two implants with peri-implant pathology [7].

Meloni included in her study 10 non-smoking patients and 5 smokers, of whom 3 smoked up to 10 cigarettes a day and 2 patients smoked more than 10 cigarettes a day [15].

Polizzi reports 2 implant failures in a patient with a smoking habit of and a consumption of 20 cigarettes per day [14].

Pozzi and Vogl exclude smokers with more than 10 cigarettes a day [19,20].

Yamada reports that the implant survival rates for smokers (13 patients, 27.1%) and non-smokers (35 patients, 72.9%) were 94.5% and 100%, respectively, with only two failures in the smoking group [19].

The results regarding the location (mandibular or maxillary, anterior or posterior) of the implant failures are shown below in Table 7. Only four studies placed implants in both the mandibular and maxillary locations and reported the failure data.

The risk of bias was considered low for all studies included in the review; the following parameters were adopted (ROBINS-I): Table 8: Due to confounding factors, bias in the selection of participants for the study, bias in the classification of interventions, bias due to deviations from intended interventions, bias due to missing data, bias in the measurement of outcomes, and bias in the selection of the reported result, for each parameter, the evaluation could be low risk, moderate risk, serious risk, or critical risk.

## 4. Discussion

This review reports short- and medium-term observations in which the stability of the marginal bone level and the high cumulative survival rate of the implant over such a period are reassuring and confirm the results already reported in the literature [23,24].

The results confirm that, if the guided surgery protocols are strictly followed and the surgery is performed with care, avoiding the displacement of the surgical guide, it is possible to adapt the prosthesis in a predictable way; in fact, as reported by the study by Marra et al. (2013) [18], the key point is the accurate positioning of the guide at the beginning of the surgery, and therefore the use of more than 3 anchor pins is recommended to improve the stabilization of the surgical guide [15,18]; furthermore, to minimize the risk of inaccurate implant placement, the majority of local anesthesia, especially on the palatal side, should be administered after fixation of the guide to minimize anesthetic-induced soft tissue deformation local.

According to van Steenberghe (1997) [25], a “successful implant” is one that:✓does not cause allergies, toxic, or infectious reactions;✓offers anchorage for the prosthesis;✓does not show any signs of fracture or flexion;✓does not show any mobility when tested through movements obtained with hand tools;✓shows no sign of radiolucency on an intraoral radiograph using a parallel beam technique perpendicular to the implant surface.

A “surviving implant” is one that remains in the bone, is stable, and is functionally successful, even if other criteria for implant success are not met. They can be defined as prosthetic successes when the prosthesis has structural stability requirements associated with effective functionality; implants are considered failed if they are removed, fractured beyond repair, or cannot be classified as surviving or successful [14].

In the study by Lopes et al. (2015), 2 early and one late failure occurred in 3 different patients; in one case the implant was lost because the patient was a strong bruxist, in another case an implant did not achieve adequate osseointegration, and in the late failure (after 45 months), the reason was due to an overload of the implant [7].

In other studies, such as that of Marra et al. (2013) [18], one failure occurred after 2 years due to peri-implantitis.

The cumulative survival of tilted and axial implants in Pozzi’s study is similar [19]; failures of two tilted and one axial implant were recorded in the same patient 4 months after loading. In this patient, implant stability was compromised due to poor bone quality, and the period of osseointegration was influenced by parafunctional habits and the debonding of the provisional prostheses. Thus, high initial stability is required when immediate or early function is desired [26]. Primary stability of the implant is provided by bone anchoring and rigid attachment to the provisional prosthesis [27].

In the study by Vogl et al. (2015) [20], no implant failures occurred, resulting in a 100% survival rate; in the same study, 3 implants did not achieve primary stability; therefore, they were not recognized as failures and were not included in the qualitative analysis.

The high survival rate observed in the various studies could be attributed to three main factors [21]:Positioning the implants as equidistant as possible and limiting the number of intermediate elements to less than two to avoid micro-movements;Adequate splinting of the implants with the provisional restoration;Avoid excessive occlusal loading by limiting occlusal contacts in the anterior six teeth and instructing patients to avoid hard foods for up to 8 weeks after the insertion of the provisional restoration.

Implant survival rates could be attributed to ideal prosthetic placement with benefits for proper oral hygiene and well-planned inter-implant distances; however, they could also be attributed to case selection; for example, in Derksen’s study implants were inserted only in the posterior region; and in the same study tissue level implants were used, which demonstrated high survival rates [17].

In a recent systematic review of the literature, the precision of the guided implantology system was highlighted, with an average error in the vertical direction of 0.5 mm and in the horizontal direction of 1.2 mm [28].

Additionally, guided implant placement is more expensive than conventional placement due to the cost of software, prosthesis duplication, a CT scan of the patient’s prosthesis, a surgical template, and planning time. However, the whole procedure from the stage surgery to the final prosthetic restoration is less invasive than the standard protocol with reduced time and minimal discomfort for the patient, and for this reason all patients stated that they could have undergone the same procedures again despite the costs [15].

The use of these technologies can bring advantages for both patients and dentists. Indeed, due to its relatively high level of accuracy, as documented in several studies [29,30], the risk of damage to the anatomical structures is minimized because the use of the three-dimensional planning system and guided surgery allows the residual bone volume to be maintained. Furthermore, surgical operating times are reduced with less discomfort and post-operative complications such as pain and swelling for the patient.

The use of implant planning software is advantageous: it is possible to place the implants practically before surgery, finding a correct position not only in the healed sites but also in post-extraction sites by anticipating the dynamic bone-level changes that occur due to a natural healing process after tooth extraction [16].

These procedures and their benefits are well established when patients are edentulous in the area to be treated, but in the event that a patient needs implants when there are still teeth to be extracted, there can be problems in diagnosing, planning, and undergoing treatment. A prerequisite before CT scan data acquisition is that the patient should have fully healed ridges, a condition not present immediately after tooth extraction. If the CT scan is performed shortly after the extraction of the remaining teeth, the bone remodeling that occurs during the early stages of healing will influence the fit of the surgical template in the patient’s mouth, resulting in a clinical situation where the volume and bone contour differ from those seen on the computer during virtual surgery and it may be impossible to place the implants correctly.

All of this requires the patient to wear a removable transitional prosthesis for 6–12 months until bone remodeling occurs to achieve a clinically successful procedure. In the study by Polizzi and Cantoni, 2015 [14], a technique was used for the fabrication of a multi-unit radiological template (composed of 2 parts: a basal part, which has holes for the teeth to be extracted, and a part obtained by waxing up the same teeth; therefore, a first scan of the patient wearing the base part of the guide is made, then a second scan is made with the 2 assembled parts of the surgical guide only), which gives patients further tangible benefits: it allows the patient to keep the teeth (which are intended for extraction) during the diagnostic phase until the day of surgery and allows prosthetically guided virtual planning of implant placement, regardless of the position of the teeth to be extracted, avoiding the need for a temporary removable prosthesis. This is the main advantage for the patient and the main advantage for the feasibility of the treatment;

In fact, the absence of the transition phase with the removal of the prosthesis means that the buccal bone cortex at the extraction sites can be kept intact, allowing implant placement, which would otherwise be impossible after resorption has taken place. Furthermore, the psychological implications and patient benefits are substantial when the prosthesis can be delivered without the transient prosthetic period.

Further, at the consensus conference on immediate loading held in Barcelona in 2002 (Rocci et al., 2003 [31]), it was stated that the minimum implant length should be 10 mm, regardless of implant diameter or design. If the implant is placed in a post-extraction site, only the most apical portion will be in the bone. To compensate, longer implants and greater numbers should be used [16].

A crucial factor is the correct selection of the macrostructure of the fixture. Despite the tendency to use tapered implants, parallel-walled implants were used in Ciabattoni’s study as they are easier to place by guided insertion, especially when the quality and quantity of residual bone are good. In cases where the density and effective residual bone volume are low, such as in extraction sites, implants with a more aggressive macro-design that take advantage of the self-tapping capacity of the fixtures are to be preferred [16].

The relatively low mean marginal bone loss observed at the extraction sites can be attributed to the flapless technique for computer-guided implant placement, which keeps the circumferential gingival fibers intact and ensures less disruption of the blood vessels feeding the buccal bone cortex.

The flapless technique was used in all studies and is advantageous, as immediate implant placement in post-extraction sites without incisions or flap creation has been shown to ensure ideal peri-implant tissue healing while minimizing crestal bone loss with preservation of bone and gingival appearance [32]. When the buccal cortex is preserved, the gingival contour can be maintained, and an immediate flapless implant placement protocol can be performed, despite the presence of endodontic or periodontal lesions [33,34,35].

The success of implant surgery is not only related to the position of the implant in the bone but also to the surrounding soft tissue. Small flap elevation (used in the studies by Pozzi et al. [19] and Derksen et al. [17]) in cases of limited keratinized tissue width appears to be beneficial for the preservation of the surrounding bone [36]. Flapless-guided surgery should only be performed if a sufficient amount of keratinized tissue is available. Conversely, freehand flapless surgery without guidance increases the risk of implant malposition and subsequent apical bone dehiscence or perforation [37,38]. This suggests that if flapless surgery is desired, computer guidance during implant site preparation and implant placement is advantageous [39].

Sanna et al. (2007) reported significant differences between smokers and non-smokers regarding marginal bone loss with flapless guided surgery: after a mean follow-up of 2.2 years, the reported marginal bone loss for smokers and non-smokers was 2.6 mm and 1.2 mm, respectively [40]. The study by Lopes et al. (2015) included two smokers, and the results followed the same pattern, with higher bone loss after 3 and 5 years for these patients and implant failure in one of these patients. Additionally, there were two implants with peri-implant pathology, which was addressed through non-surgical and surgical therapies [22].

The three-dimensional planning of the software plays a key role in evaluating the integrity and thickness of the vestibular cortex, while the virtual positioning system allows us to manage the correct orientation of the fixtures by evaluating the correct relationship between the implant diameter and the distance between the implants and buccal bone cortex [41,42].

In the study by Pozzi et al. (2012) [19], the clinical and radiographic parameters of axial and inclined implants were studied The 2 groups were clinically equal because the differences in the marginal bone loss of each study group were less than 0.1 mm. According to Pozzi et al. [19], marginal bone loss was slightly affected by the inclination of the implants; when the angled implants were fixed to the axial implants, they showed a similar type of bone response to that reported in previous studies [26,43].

Unlike axial implants, tilted implants are allocated and stabilized with a greater surface area in the cortical bone. The positioning of the tilted implants requires greater competence and skill on the part of the executor, especially if there is a greater degree of invasiveness due to the involvement of the maxillary sinus with a perforation. The use of a straight probe is recommended for the evaluation of the maxillary sinus edges. The reduction of surgical invasiveness and postoperative symptoms can be achieved with guided implant surgery, with the achievement of more predictable results [19].

Bone resorption around dental implants is reduced by applying a combination of a minimally invasive guided approach and the biomedical characteristics of CAD/CAM abutments. The application of guided implant surgery using computer programs allows the oral surgeon to place tilted dental implants in dense cortical bone structures with a tricortical anchorage, reducing or even eliminating the application of bone substitute biomaterials and bone grafts. Naturally, a fundamental prerequisite for the use of this technique is the presence of a suitable bone volume in the posterior and anterior walls of the maxillary sinus.

Several studies [44,45] have shown that angled implants can increase stress on the surrounding bone. However, when the implant is fixed to multiple implant-fixed prostheses, the stiffness of the prosthetic structure should reduce the stress.

The study results of this review confirm that immediate loading is a reasonable alternative to the classic two-stage procedure, and involves flapless surgery and mild postoperative discomfort [15].

In the study by Vogl et al. (2015), the high immediate occlusal load in the posterior region of the partially edentulous mandibles did not appear to have influenced the osseointegration of the implants [20].

Östman et al. (2012) reported a 99% cumulative survival rate after 1 year of immediate occlusal loading of 139 implants, with a mean bone resorption of 1.01 mm [46]. Cannizzaro et al. (2011) reported a survival rate of 94% and marginal bone resorption between 0.24 and 0.33 mm after one year of immediate occlusal loading [47].

It is essential to reduce micromovements at the bone-implant interface to achieve and maintain osseointegration in immediate loading procedures [48].

According to a review of complications in guided surgery by Hultin et al. (2012), intraoperative complications include occlusal index maladjustment caused by maxillomandibular registration errors, difficulty in using surgical templates due to insufficient mouth opening, guide maladjustment, insufficient primary stability of the implants resulting from soft bone tissue, and guide fracture surgery [49].

Among the main post-operative surgical complications reported, excluding failures due to lack of osseointegration, we have the lack of keratinization in the implant area as reported by Derksen [17], moreover, Polizzi reports spontaneous bleeding and exudation in post-extraction sites with the presence of gingivitis in 3 patients [14]. Other possible postoperative complications of this procedure include pain, discomfort, swelling, bone necrosis caused by inadequate drilling, and bone resorption resulting from excessive compression [21].

Concerning the position of the implant failures, we have only a number of 4 studies that report the data both on the mandibular and maxillary positioning, from these 4 studies, the total number of maxillary implants was 508 with a number of failures equal to 13, while the mandibular ones were 324 with 4 failures. It was decided not to perform a meta-analysis of the data for the small number of potentially included studies; the data on the placement of the implants on the anterior and posterior sectors for the included studies are incomplete, and the reported failures would indicate a homogeneous distribution of the failures between the 2 sectors (5 posterior and 4 anterior). Further evaluations would be useless due to the lack of reported data in studies.

Frequently reported restorative complications during the insertion of prefabricated dentures include the need for extensive occlusal adjustment, maladjustment of the denture, and pain during restorative procedures. Komiyama et al. (2008) reported the mismatch between abutments and dentures and the need for extensive occlusal adjustments in approximately 10% of patients in whom immediate loading was performed in edentulous jaws following computer-assisted surgery procedures [50,51].

Screw loosening, implant loss, and fracture of provisional prostheses have been observed as postoperative complications [21]. Screw loosening was observed in 10 implants placed more posteriorly, angled, and fixed to angled abutments. Two of the four lost implants were placed in the posterior region (first molars), with inclination. Occlusal overload in posterior regions where forces are very intense could be the main cause of these complications. The other 2 implant prosthetic failures developed in the mandible in the anterior regions (lateral incisor) with fractures of the provisional teeth; the implants were not inclined, and these failures are also attributable to excessive occlusal forces due to the absence of teeth in the posterior regions.

The imperfect adaptation of the provisional prosthesis found in 2 studies (Meloni et al., 2010 [15]; Vogl et al., 2015 [20]) could have been caused by a deviation of the implant, which in turn, as reported by Meloni in his study, may have been caused by the fracture of the surgical guide during implant placement [15].

The high incidence of mechanical complications observed in the study by Lopes et al. is related to the high incidence of bruxist patients; in one study, bruxism was considered a risk factor for mechanical complications when complete rehabilitations of edentulous saddles were performed [22,52].

## 5. Conclusions

On the basis of the studies selected for this review, which relate to the scientific literature of the last 10 years, it can be seen that the implant survival obtained with the use of guided implant surgery shows high percentages.

In addition, many recorded failures occurred early, due to a lack of osseointegration, and the variables that come into play in the survival of the implants are many.

Further studies should be conducted, with a longer-term follow-up, to understand what possible further evaluations can be carried out in order to have not only a higher survival rate but also a more predictable success.

## Figures and Tables

**Figure 1 jcm-12-01490-f001:**
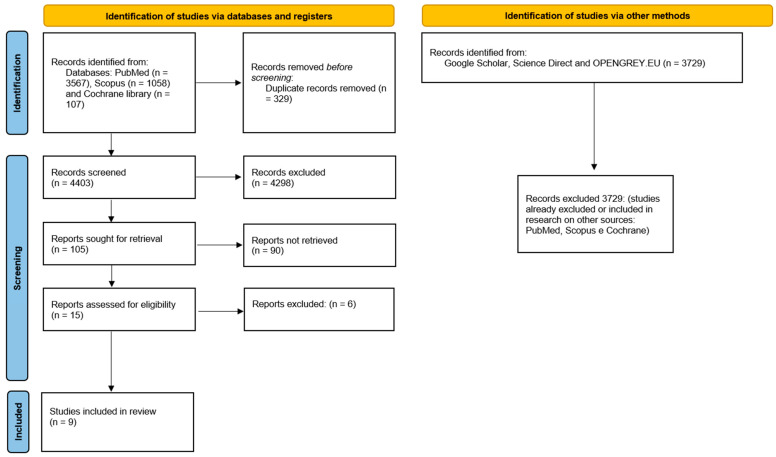
Flowchart of the article selection process.

**Table 1 jcm-12-01490-t001:** Description of studies by study type, number of patients, age, number of implants, diameter and length of implants, and duration of follow-up, (/) data not reported.

First Autor, Data	Type of Study	Number Patient (Female/Male)	Age (Average)	Number Implant	Diameter/Length Implants (mm)	Follow up
Ciabattoni et al., 2017 [16];	Prospective	32 (23/9)	44–73 (59.5)	197 in post-extraction sites; 88 in healed sites	/	3 years
Derksen et al., 2019 [17]	Prospective	66 (36/30)	20–73 (52.4)	145	3.3-4.8/8-10-12	1 and 2 years
Lopes et al., 2015 [7]	Prospective	23 (13/10)	34–70 (55.4)	92	4/8-10-11.5-13-15-18	1, 3 and 5 years
Marra et al., 2013 [18]	Prospective	30 (18/12)	/	312	3.3-3.75-4/8.5-18	3 years
Meloni et al., 2010 [15]	Retrospective	15 (10/5)	40–70 (52)	90	4.3-5/10-13	18 months
Polizzi and Cantoni, 2015 [14]	Retrospective	27 (20/7)	34–71 (55.8)	92 in healed sites; 68 in post-extraction sites	/	5 years
Pozzi et al., 2012 [19]	Prospective	27 (12/15)	38–77 (54.18)	39 axials; 42 tilted	2.8-3.2-3.-8/14.4 (axials)-16.1 (tilted)	43.4 months
Vogl et al., 2015 [20]	Prospective	20 (13/7) divided into 2 groups: 9 immediate occlusal loading; 10 non-occlusal load	33-70 (54)	21 (with immediate occlusal loading) 31 (without occlusal load)	/	1 year
Yamada et al., 2015 [21]	Prospective	48 (22/26)	34–74 (56)	278	3.5-4.3-5/8.5-18	1 year

**Table 2 jcm-12-01490-t002:** Description of the studies by state of the dental arch, condition of the implant site, type of surgery, loading protocol, and placement time of the definitive restoration, (/) data not reported.

First Autor, Data	State of the Dental Arch	Condition of the Implant Site	Flap/Miniflap/Flapless	Protocol (Immediate Loading, Conventional)	Final Restoration (Months)
Ciabattoni et al., 2017 [16];	Total edentulism	Post-extraction and healed	Flapless	Immediate	/
Derksen et al., 2019 [17]	Partial edentulism	Healed	Flapless (34) and miniflap (111)	Conventional	4
Lopes et al., 2015 [7]	Total edentulism	Healed	flapless	Immediate	4
Marra et al., 2013 [18]	Total edentulism	Healed	Flapless	Immediate	4–6
Meloni et al., 2010 [15]	Total edentulism	Healed	Flapless	Immediate	6
Polizzi and Cantoni, 2015 [14]	Partial/total edentulism	Post-extraction and healed	Flapless	Immediate	6
Pozzi et al., 2012 [19]	Partial edentulism	Healed	Flapless; miniflap	Immediate	6
Vogl et al., 2015 [20]	Partial edentulism	Healed	Flapless	Immediate	6–8
Yamada et al., 2015 [21]	Total edentulism	Healed	Flapless	Immediate	4–7

**Table 3 jcm-12-01490-t003:** Number of implant failures, implant survival rate, mean marginal bone loss, and implant-prosthetic complications, (/) data not reported.

First Autor, Data	Failure	Implant Survival Rate	Medium Marginal Bone Loss	Implant-Prosthetic Complications
Ciabattoni et al., 2017 [16];	Early: 5 in extraction sites (after 6 months) late: 2 in healed sites (after 2 and 3 years)	97.54%	Mm	Postoperative: implant loss (7).
Derksen et al., 2019 [17]	1 early failure (6 weeks)	99.3%	/	Intraoperative: Inability to use a drill due to limited mouth opening (1), vestibular dehiscence after osteotomy (1).
Lopes et al., 2015 [7]	Early: 2 after 5 months Late: 1 after 3 years	96.6%	1.9 mm	Postoperative: Abutment screw loosening (2), definitive prosthesis fracture (7), implant loss (3).
Marra et al., 2013 [18]	Early: 3 in the first 3 months, 2 after 6 months Late: 1 after 1 year, 1 after 2 years	97.9%	Mm	Postoperative: Prosthesis Fracture (9), implant loss (6).
Meloni et al., 2010 [15]	Early: 2 after 6 months	97.8%	1.6 mm	Postoperative: Implant loss (2), imperfect fit of the provisional prosthesis (2), fracture of the provisional prosthesis (1).
Polizzi and Cantoni, 2015 [14]	Early: 2 after 6 months (post-extraction site) Late: 2 in healed sites (after 2 years)	97.33%	1.39 mm	Postoperative: Implant loss (4).
Pozzi et al., 2012 [19]	Early: 1 axial, 2 tilted, after 4 months in the same patient	96.3%	0.6 mm	Postoperative: Implant loss (3).
Vogl et al., 2015 [20]	0	100%	0.4 ± 0.5 mm	Intraoperative: Insufficient bone quantity (1), insufficient primary stability (3); Postoperative: Imperfect fit of temporary prostheses (3), occlusal adjustments required (2), fractures (2), abutment screw loosening (3).
Yamada et al., 2015 [21]	Early: 2 in one patient (2 weeks); 2 in a patient (1 month)	98.6%	0.32 mm	Postoperative: Abutment screw loosening (10), temporary fracture prosthesis (1), implant mobility (4).

**Table 4 jcm-12-01490-t004:** Early and late failure rate in each study and overall failure rate, implant level, (/) data not reported.

First Autor, Data	N. Implant	Early Failure Rate	Late Failure Rate
Ciabattoni et al., 2017 [16];	285	1.75%	0.70%
Derksen et al., 2019 [17]	145	0.69%	0%
Lopes et al., 2015 [7]	92	2.17%	1.09%
Marra et al., 2013 [18]	312	1.60%	0.64%
Meloni et al., 2010 [15]	90	2.22%	0%
Vogl et al., 2015 [20]	52	0%	0%
Polizzi and Cantoni, 2015 [14]	160	1.25%	1.25%
Pozzi et al., 2012 [19]	81	3.70%	0%
Yamada et al., 2015 [21]	278	1.44%	0%
Total failure rate		1.60%	0.47%

**Table 5 jcm-12-01490-t005:** Early and late failure rate in each study and overall failure rate, patient level.

First Autor, Data	N. Patient	Early Failure Rate	Late Failure Rate
Derksen et al., 2019 [17]	66	1.51%	0%
Lopes et al., 2015 [7]	23	8.69%	4.35%
Meloni et al., 2010 [15]	15	13.33%	0%
Polizzi and Cantoni, 2015 [14]	27	3.70%	3.70%
Pozzi et al., 2012 [19]	27	3.70%	0%
Vogl et al., 2015 [20]	20	0%	0%
Yamada et al., 2015 [21]	48	4.16%	0%
Total failure rate		3.98%	0.88%

**Table 6 jcm-12-01490-t006:** Biological complications and presence of smokers in the study groups. the inclusion criteria regarding smoking are reported with the number of smoking patients and the number of failed implants in smokers, in addition the number of biological complications such as failed osseointegration and the presence of peri-implantitis are reported (^1^ the term heavy smokers is reported by Lopez in his study without quantitatively specifying the number of cigarettes, \ data not reported).

First Autor, Data	Type of Study	Exclusion Criteria Adopted in Relation to Cigarette Smoking	Number Patient	Number Implant	Number of Smoking Patients	Number of Failed Implants in Smokers	Biological Complications	
							Failures Due to Lack of Osseointegration (within 6 Months)	Peri-Implant Pathology, Periimplantitis
Ciabattoni et al., 2017 [16];	Prospective	more than 10 cigarettes/day	32	285	\	\	5	\
Derksen et al., 2019 [17]	Prospective	smokers were not excluded	66	145	\	1	1	\
Lopes et al., 2015 [22]	Prospective	smokers were not excluded	23	92	2 (heavy smokers) ^1^	\	2	2 peri-implant pathology
Marra et al., 2013 [18]	Prospective	smokers were not excluded	30	312	\	\	3	1 implant lost after 2 years (periimplantitis)
Meloni et al., 2010 [15]	Retrospective	smokers were not excluded	15	90	5 (3 patients smoked up to 10 cigarettes a day, 2 smoked more than 10 cigarettes per day)	\	2	2 periimplantitis
Polizzi and Cantoni, 2015 [14]	Retrospective	Smoker patients (≤20 cigarettes/day) were not excluded	27	160	\	2	2	24 implants in 3 patients (periimplantitis)
Pozzi et al., 2012 [19]	Prospective	more than 10 cigarettes/day	27	81	\	\	3	\
Vogl et al., 2015 [20]	Prospective	more than 10 cigarettes/day	20	52	\	\	0	3 mucositis
Yamada et al., 2015 [21]	Prospective	smokers were not excluded	48	278	13	4	4	\

**Table 7 jcm-12-01490-t007:** Number of failures in relation to the position of the dental implants, Anterior sinus wall implants (ASW), posterior sinus wall implants (PSW), Right (R), Left (L), data not reported (/).

First Autor, Data	Position	Implant Failure	Anterior Position (Number of Implant Failures)	Posterior Position (Number of Implant Failures)	Maxilla (Number of Implant Failures)	Mandibular (Number of Implant Failures)
Ciabattoni et al., 2017 [16];	Maxilla\mandible	7	\	\	5\193	2\90
Derksen et al., 2019 [17]	Maxilla\mandible	1	\	\	66	1\79
Lopes et al., 2015 [7]	Maxilla\mandible	3	Maxilla (lateral incisor)	2Maxilla (first molar, second premolar)	3\72	0\20
Marra et al., 2013 [18]	Maxilla\mandible	6	\	\	5\177	1\135
Meloni et al., 2010 [15]	Maxilla	2	\	\	2	\
Polizzi and Cantoni, 2015 [14]	Maxilla	4	\	\	4	\
Pozzi et al., 2012 [19]	Maxilla	3	2 ASW	1 PSW	3	\
Vogl et al., 2015 [20]	Mandible	\	\	\	\	\
Yamada et al., 2015 [21]	Maxilla	4	2 maxilla (incisive lateral R and incisive lateral L)	2 maxilla (first molar R and first molar L)	4	\

**Table 8 jcm-12-01490-t008:** Risk of bias; Low risk +, Moderate risk -, Serious risk x, Critical risk!

	Bias Due to Confounding	Bias in Selection of Participants into the Study	Bias in Classification of Interventions	Bias Due to Deviations from Intended Interventions	Bias Due to Missing Data	Bias in Measurement of Outcomes	Bias in Selection of the Reported Result	Overall Bias
Ciabattoni et al., 2017 [16];	+	+	+	+	+	+	+	+
Derksen et al., 2019 [17]	+	+	+	+	+	+	+	+
Lopes et al., 2015 [7]	+	+	+	+	+	+	+	+
Marra et al., 2013 [18]	+	+	+	+	+	+	+	+
Meloni et al., 2010 [15]	+	+	+	+	-	+	+	+
Polizzi and Cantoni, 2015 [14]	+	+	+	+	+	+	+	+
Pozzi et al., 2012 [19]	+	+	+	+	+	+	+	+
Vogl et al., 2015 [20]	+	+	+	+	+	+	+	+
Yamada et al., 2015 [21]	+	+	+	+	+	+	+	+

## Data Availability

Not applicable.

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
