# Peer review of "Guided Dental Implant Surgery: Systematic Review"

_jcm, 2023, doi:10.3390/jcm12041490_

Round 1
Reviewer 1 Report
Guided dental implant Surgery: Systematic Review.
J. Clin. Med. 2022, 11, x. https://doi.org/10.3390/
The present systematic review appears to be interesting for the clinical relevance of the topic in implant dentistry, guided implant surgery. Some studies had showed flappless guided surgery indications for implant treatment. The purpose of this review was to evaluate survival rates, early and late failure rate, peri-implant bone remodeling and any implant-prosthetic complications related to implants inserted using digitally designed surgical guides in the scientific literature of the last years.
Introduction. The introduction is appropriate according to the topic addressed, but it should provide more data about this field:
Velasco-Ortega E, Jiménez-Guerra A, Ortiz-Garcia I, Moreno-Muñoz J, Núñez-Márquez E, Cabanillas-Balsera D, López-López J, Monsalve-Guil L. Immediate loading of implants placed by guided surgery in geriatric edentulous mandible patients. Int J Environ Res Public Health 2021,18, 4125. DOI:10.3390/ijerph18084125
Moreover, the first and second paragraphs must include some references.
The authors must include in the purpose of the review the prevalence of biologic complications as peri-implantitis. This complication is very important for the long-term success of implant treatment.
Material and methods. The methodology is registered and widely described. However, it needs some improvements.
The keywords and/or terms ("Surgery, Computer-Assisted" AND "implant survival") used do not correspond to the objectives set out in this systematic review. Terms related to early and late failure rate, bone remodeling for implant and any implant prosthetic complications should be included.
Search terms should be reviewed: such as those related to embryos should be eliminated because they have no dental relationship; o mortality, mortality or surviving are not part of the implant survival issue.
Due to the systematic nature of the review, the study selection process must be detailed according to the application of the inclusion and exclusion criteria, as well as it is recommended to carry out this process by several researchers (minimum two, identify who, and a third investigator for possible disagreements).
Data extraction is well described.
Results. Nine articles are included in the systematic review. For the rehabilitation of the totally or partially edentulous arches, a variable number of implants was inserted. The follow up of the studies ranges from a minimum of 1 year to a maximum of 5 years.
In almost all the studies analyzed, guided surgery was used for the rehabilitation of totally edentulous arches; the implants were inserted both in post-extraction sites and in healed sites.
Flapless surgery was used in all studies, although the use of open flap surgery was reported in one study and miniflaps in another study. Immediate loading was the most frequent protocol, except one study which used conventional loading.
Also, this section shows the number of implant failures, survival rate, mean marginal bone loss, and implant-prosthetic complications. Both early failures as early as 2 weeks after insertion and late failures after 3 years were found. The mean marginal bone loss ranges from a minimum value of 0.32 mm after a 1 year follow-up to a maximum value of 1.9 mm.
The paragrah included between 188 to 205 lines is not correct. The prosthetic implant complications must be referred to 195 to 205 lines. The previous paragraph (Intraoperative) shows surgery complications.
The authors must include in this section of the review, the data of biologic complications as peri-implantitis.
Discussion. This section shows a well analysis of the results of the study. However, some considerations are necessary.
Smoking habits is not included in the results of the work, but in the discussion are present in 370-377 lines.
The authors show some aspects of discussion of prosthetic complications (426 -449 lines), but not shows a discussion of postoperative surgical complications (423-425 lines).
Conclusions. This section must be related with the objectives of the paper. The paragraphs 3 and 4 shows conclusions non related with the aim of the work.
References. During last years, there are many scientific studies related with computer guided implant dentistry. The authors must update this section.
Author Response
Reviewer
Guided dental implant Surgery: Systematic Review.
- Clin. Med. 2022, 11, x. https://doi.org/10.3390/
The present systematic review appears to be interesting for the clinical relevance of the topic in implant dentistry, guided implant surgery. Some studies had showed flappless guided surgery indications for implant treatment. The purpose of this review was to evaluate survival rates, early and late failure rate, peri-implant bone remodeling and any implant-prosthetic complications related to implants inserted using digitally designed surgical guides in the scientific literature of the last years.
Answer
Thank you for reviewing the manuscript. Your suggestions and comments have been helpful in improving the article
Reviewer
Introduction. The introduction is appropriate according to the topic addressed, but it should provide more data about this field:
Velasco-Ortega E, Jiménez-Guerra A, Ortiz-Garcia I, Moreno-Muñoz J, Núñez-Márquez E, Cabanillas-Balsera D, López-López J, Monsalve-Guil L. Immediate loading of implants placed by guided surgery in geriatric edentulous mandible patients. Int J Environ Res Public Health 2021,18, 4125. DOI:10.3390/ijerph18084125
Velasco-Ortega E, Jiménez-Guerra A, Ortiz-Garcia I, Matos Garrido N, Moreno-Muñoz J, Núñez-Márquez E, Rondón-Romero JL, Cabanillas-Balsera D, López-López J, Monsalve-Guil L. Implant treatment by guided surgery supporting overdentures in edentulous mandible patients. Int J Environ Res Public Health 2021, 18, 11836. https://doi.org/ 10.3390/ijerph182211836
Moreover, the first and second paragraphs must include some references.
The authors must include in the purpose of the review the prevalence of biologic complications as peri-implantitis. This complication is very important for the long-term success of implant treatment.
Answer
Introduction
new information has been added as suggested in the introduction and bibliographic references have been added to the paragraphs, furthermore the prevalence of biological complications such as peri-implantitis have been added in the objectives of the review. text changes are shown below (yellow)
Velasco-Ortega et al. reports in a very recent study on guided implantology conducted on 22 geriatric patients with 198 implants positioned in the mandible reports a cumulative survival rate for all implants was 97.5%. The number of implants lost was 5 and the post-operative complications involved 10 patients with peri-implantitis and 6 mechanical prosthodontic complications[1]; Furthermore, these data are partially confirmed by a subsequent study conducted by the same research group on a smaller number of patients and the implants with prosthetic overdentures positioned, reporting a 100% survival rate after an average follow-up period of 44 months [2-4].
Based on the data in the literature it can be seen that guided flapless surgery is comparable to freehand surgery in terms of survival, marginal bone remodeling and peri-implant variables [5].
The aim of this work is to evaluate survival rates, early and late failure rate, peri-implant bone remodeling and any implant-prosthetic complications related to implants inserted using digitally designed surgical guides in the scientific literature of the last 10 years, at in order to evaluate the possibilities of applying guided surgery in implantology clinical procedures, moreover, the prevalence of biological complications such as peri-implantitis will also be evaluated.
Reviewer
Material and methods. The methodology is registered and widely described. However, it needs some improvements.
The keywords and/or terms ("Surgery, Computer-Assisted" AND "implant survival") used do not correspond to the objectives set out in this systematic review. Terms related to early and late failure rate, bone remodeling for implant and any implant prosthetic complications should be included.
Search terms should be reviewed: such as those related to embryos should be eliminated because they have no dental relationship; o mortality, mortality or surviving are not part of the implant survival issue.
Due to the systematic nature of the review, the study selection process must be detailed according to the application of the inclusion and exclusion criteria, as well as it is recommended to carry out this process by several researchers (minimum two, identify who, and a third investigator for possible disagreements).
Data extraction is well described.
Answer
The irrelevant keywords have been eliminated and the suggested terms have been added and the database search has been re-executed by developing a new flow chart, furthermore the inclusion and exclusion criteria have been reformulated in a clearer way and the selection process has been better described and choice of studies to include (with 2 reviewers and a third for equivocal situations). text changes are shown below (yellow)
2.2. Information sources, Search
Between 1 November 2022 and 7 November, a systematic search was carried out independently by 2 reviewers in the PubMed, Scopus and Cochrane Library databases with the implementation of the Science Direct and google scholar databases for the analysis of gray literature; for the purpose of minimizing Publication Bias.
The keywords used were the following: Computer-Assisted, early and late failure rate, bone remodeling. dental implant, prosthetic complications.
In particular, the terms used in detail on the PubMed search engine are the following:
Search: Computer-Assisted AND (early late failure rate OR bone remodeling OR dental implant OR prosthetic complications) Sort by: Most Recent: "Computer-Assisted"[All Fields] AND (("early"[All Fields] AND "late"[All Fields] AND ("failure"[All Fields] OR "failures"[All Fields]) AND ("j rehabil assist technol eng"[Journal] OR "rate"[All Fields])) OR ("bone remodelling"[All Fields] OR "bone remodeling"[MeSH Terms] OR ("bone"[All Fields] AND "remodeling"[All Fields]) OR "bone remodeling"[All Fields]) OR ("dental implants"[MeSH Terms] OR ("dental"[All Fields] AND "implants"[All Fields]) OR "dental implants"[All Fields] OR ("dental"[All Fields] AND "implant"[All Fields]) OR "dental implant"[All Fields]) OR (("prosthetic"[All Fields] OR "prosthetically"[All Fields] OR "prosthetics"[All Fields]) AND ("complicances"[All Fields] OR "complicate"[All Fields] OR "complicated"[All Fields] OR "complicates"[All Fields] OR "complicating"[All Fields] OR "complication"[All Fields] OR "complication s"[All Fields] OR "complications"[MeSH Subheading] OR "complications"[All Fields])))
Translations: failure: "failure"[All Fields] OR "failures"[All Fields]. Rate: "J Rehabil Assist Technol Eng"[Journal:__jid101671667] OR "rate"[All Fields]. Bone remodeling: "bone remodelling"[All Fields] OR "bone remodeling"[MeSH Terms] OR ("bone"[All Fields] AND "remodeling"[All Fields]) OR "bone remodeling"[All Fields]. dental implant: "dental implants"[MeSH Terms] OR ("dental"[All Fields] AND "implants"[All Fields]) OR "dental implants"[All Fields] OR ("dental"[All Fields] AND "implant"[All Fields]) OR "dental implant"[All Fields]. prosthetic: "prosthetic"[All Fields] OR "prosthetically"[All Fields] OR "prosthetics"[All Fields]. complications: "complicances"[All Fields] OR "complicate"[All Fields] OR "complicated"[All Fields] OR "complicates"[All Fields] OR "complicating"[All Fields] OR "complication"[All Fields] OR "complication's"[All Fields] OR "complications"[Subheading] OR "complications"[All Fields].
Furthermore, a last update of the records and keywords used was carried out on the databases on 16 December 2022.
2.3. Selection of studies
The selection of the studies was conducted by 2 authors independently with a 3 with the task of evaluating the situations of doubt and conflict between the 2 authors. The selection of the studies and their relative inclusion was based on the application of the inclusion and exclusion criteria
In order to assess the suitability of the studies, all the titles and abstracts of the publications generated by the research were consulted. The full text of the articles was retrieved in studies that appeared to meet the screening criteria and in studies in which the title and abstract did not give sufficient information to firmly decide whether to include the study or not.
Inclusion criteria: All guided surgery studies in the implantology field were included, in which there are data regarding implant survival, marginal bone resorption, intra and postoperative complications; the selected studies include observational, prospective, retrospective studies, RCTs, and multicenter studies.
Exclusion criteria: Animal studies, guided surgery studies involving other anatomical areas, systematic reviews and meta-analyzes were excluded, case report, in vitro study, studies without abstracts in English.
Reviewer
Results. Nine articles are included in the systematic review. For the rehabilitation of the totally or partially edentulous arches, a variable number of implants was inserted. The follow up of the studies ranges from a minimum of 1 year to a maximum of 5 years.
In almost all the studies analyzed, guided surgery was used for the rehabilitation of totally edentulous arches; the implants were inserted both in post-extraction sites and in healed sites.
Flapless surgery was used in all studies, although the use of open flap surgery was reported in one study and miniflaps in another study. Immediate loading was the most frequent protocol, except one study which used conventional loading.
Also, this section shows the number of implant failures, survival rate, mean marginal bone loss, and implant-prosthetic complications. Both early failures as early as 2 weeks after insertion and late failures after 3 years were found. The mean marginal bone loss ranges from a minimum value of 0.32 mm after a 1 year follow-up to a maximum value of 1.9 mm.
The paragrah included between 188 to 205 lines is not correct. The prosthetic implant complications must be referred to 195 to 205 lines. The previous paragraph (Intraoperative) shows surgery complications.
The authors must include in this section of the review, the data of biologic complications as peri-implantitis.
Answer
The errors on lines between 188 and 205 have been corrected and modified as required. In addition, data on biological complications such as peri-implantitis were added as required. ). text changes are shown below (yellow)
The implant complications reported in the various studies are:
- Intraoperative (surgery complications), such as the impossibility of using a drill due to limited opening of the mouth of a patient, buccal bone dehiscence after osteotomy in another patient (both cases in the study by Derksen et al. 2019 [6] ), insufficient bone quantity in one patient and insufficient primary stability in 3 patients (Vogl et al., 2015 [7]; these cases were not considered in the study statistics, thus resulting in a 100% survival rate).
- Postoperative,(prosthetic implant complications) including loss of implants in almost all studies, loosening of the abutment screw in 2 cases in the study by Lopes et al., 2015[8], in 3 cases in the study by Vogl et al., 2015 [7], in 10 cases in the study by Yamada et al., 2015 [9]; fracture of the definitive prosthesis in 7 cases in the study by Lopes et al., 2015 [10], in 9 cases in the study by Marra et al., 2013 [11], in 2 cases in the study by Vogl et al., 2015 [7];, fracture of provisional prostheses in 2 cases in the study by Meloni et al., 2010 [12], in one case in the study by Yamada et al., 2015 [9]; imperfect fit of the provisional prosthesis in 2 cases in the study by Meloni et al., 2010 [12], in 3 cases in the study by Vogl et al., 2015 [7]; need for occlusal adjustments in 2 cases in the study by Vogl et al. 2015 [18].
The data on peri-implantitis and more generally on the biological complications are reported in detail in table 6, the most commonly reported post-operative biological complication which is also the final event that leads to the loss of the implant is the lack of osseointegration. The presence of peri-implantitis is reported in at least 4 studies and appears to be the main complication. An additional consideration should be made on the consumption of smoked tobacco, it should be noted that the early loss of the implant due to lack of osseointegration occurred in smoking patients in at least 3 studies.
The data related to smoking are reported in table 6 and I describe a brief summary below.
Ciabattoni reports heavy smokers with a consumption of more than 10 cigarettes a day as exclusion criteria. He also advises smokers not to smoke for at least a week after surgery[13].
Derkesen reports early mandibular implant failure in a smoker and smoking was not an exclusion criterion[6].
Lopes included two smokers who had more marginal bone loss after 3 and 5 years of follow up and further implant failure. In addition, there were two implants with peri-implant pathology [8].
Meloni included in her study 10 non-smoking patients, and 5 smokers of which 3 smoked up to 10 cigarettes a day and 2 patients smoked more than 10 cigarettes a day [12].
Polizzi reports 2 implant failures in a patient with a habit of smoking with a consumption of 20 cigarettes per day[14].
Pozzi and Vogl exclude smokers with more than 10 cigarettes a day [7,15].
Yamada reports that the implant survival rates for smokers (13 patients, 27.1%) and non-smokers (35 patients, 72.9%) were 94.5% and 100%, respectively. With only 2 failures in the smoking group[19.
Table 6. biological complications and presence of smokers in the study groups. the inclusion criteria regarding smoking are reported with the number of smoking patients and the number of failed implants in smokers, in addition the number of biological complications such as failed osseointegration and the presence of peri-implantitis are reported (1 the term heavy smokers is reported by Lopez in his study without quantitatively specifying the number of cigarettes, \ data not reported).
Reviewer
Discussion. This section shows a well analysis of the results of the study. However, some considerations are necessary.
Smoking habits is not included in the results of the work, but in the discussion are present in 370-377 lines.
The authors show some aspects of discussion of prosthetic complications (426 -449 lines), but not shows a discussion of postoperative surgical complications (423-425 lines).
Answer
a table reporting smoking habits in relation to implant failures in guided surgery was added to the results, and a further paragraph on postoperative surgical complications was added. ). text changes are shown below (yellow)
The data related to smoking are reported in table 6 and I describe a brief summary below.
Ciabattoni reports heavy smokers with a consumption of more than 10 cigarettes a day as exclusion criteria. He also advises smokers not to smoke for at least a week after surgery[13].
Derkesen reports early mandibular implant failure in a smoker and smoking was not an exclusion criterion[6].
Lopes included two smokers who had more marginal bone loss after 3 and 5 years of follow up and further implant failure. In addition, there were two implants with peri-implant pathology [8].
Meloni included in her study 10 non-smoking patients, and 5 smokers of which 3 smoked up to 10 cigarettes a day and 2 patients smoked more than 10 cigarettes a day [12].
Polizzi reports 2 implant failures in a patient with a habit of smoking with a consumption of 20 cigarettes per day[14].
Pozzi and Vogl exclude smokers with more than 10 cigarettes a day [7,15].
Yamada reports that the implant survival rates for smokers (13 patients, 27.1%) and non-smokers (35 patients, 72.9%) were 94.5% and 100%, respectively. With only 2 failures in the smoking group[19.
|
First Autor, data |
Type of Study |
Exclusion criteria adopted in relation to cigarette smoking |
Number patient |
Number implant |
Number of smoking patients |
Number of failed implants in smokers |
Biological complications |
|
|
Failures due to lack of osseointegration (within 6 months) |
Peri-implant pathology, periimplantitis. |
|||||||
|
Ciabattoni et al., 2017 [13]; |
Prospective |
more than 10 cigarettes/day |
32 |
285 |
\ |
\ |
5 |
\ |
|
Derksen et al., 2019 [6] |
Prospective |
smokers were not excluded |
66 |
145 |
\ |
1 |
1 |
\ |
|
Lopes et al., 2015 [10] |
Prospective |
smokers were not excluded |
23 |
92 |
2 (heavy smokers)1 |
\ |
2 |
2 peri-implant pathology |
|
Marra et al., 2013 [11] |
Prospective |
smokers were not excluded |
30 |
312 |
\ |
\ |
3 |
1 implant lost after 2 years (periimplantitis) |
|
Meloni et al., 2010 [12] |
Retrospective |
smokers were not excluded |
15 |
90 |
5 (3 patients smoked up to 10 cigarettes a day, 2 smoked more than 10 cigarettes per day) |
\ |
2 |
2 periimplantitis |
|
Polizzi and Cantoni, 2015 [14] |
Retrospective |
Smoker patients (≤ 20 cigarettes/day) were not excluded |
27 |
160 |
\ |
2 |
2 |
24 implants in 3 patients ( periimplantitis) |
|
Pozzi et al., 2012 [15] |
Prospective |
more than 10 cigarettes/day |
27 |
81 |
\ |
\ |
3 |
\ |
|
Vogl et al., 2015 [7] |
Prospective |
more than 10 cigarettes/day |
20
|
52 |
\ |
\ |
0 |
3 mucositis |
|
Yamada et al., 2015 [9] |
Prospective |
smokers were not excluded |
48 |
278 |
13 |
4 |
4 |
\ |
Table 6. biological complications and presence of smokers in the study groups. the inclusion criteria regarding smoking are reported with the number of smoking patients and the number of failed implants in smokers, in addition the number of biological complications such as failed osseointegration and the presence of peri-implantitis are reported (1 the term heavy smokers is reported by Lopez in his study without quantitatively specifying the number of cigarettes, \ data not reported).
The results regarding the location (mandibular or maxillary, anterior or posterior) on the implant failures are shown below in table 7. Only 4 studies placed implants in both the mandibular and maxillary locations reporting the data on the failures
Among the main post-operative surgical complications reported excluding failures due to lack of osseointegration we have the lack of keratinization in the implant area as reported by Derksen [6], moreover Polizzi reports spontaneous bleeding and exudation in post-extraction sites with the presence of gingivitis in 3 patients [14]. Other possible postoperative complications of this procedure include pain, discomfort, swelling, bone necrosis caused by inadequate drilling, and bone resorption resulting from excessive compression [9].
Reviewer
Conclusions. This section must be related with the objectives of the paper. The paragraphs 3 and 4 shows conclusions non related with the aim of the work.
Answer
The paragraphs have been deleted as requested
Reviewer
References. During last years, there are many scientific studies related with computer guided implant dentistry. The authors must update this section
Answer
Additional references have been added
- Velasco-Ortega, E.; Jiménez-Guerra, A.; Ortiz-Garcia, I.; Moreno-Muñoz, J.; Núñez-Márquez, E.; Cabanillas-Balsera, D.; López-López, J.; Monsalve-Guil, L. Immediate Loading of Implants Placed by Guided Surgery in Geriatric Edentulous Mandible Patients. Int J Environ Res Public Health 2021, 18, doi:10.3390/ijerph18084125.
- Velasco-Ortega, E.; Jiménez-Guerra, A.; Ortiz-Garcia, I.; Garrido, N.M.; Moreno-Muñoz, J.; Núñez-Márquez, E.; Rondón-Romero, J.L.; Cabanillas-Balsera, D.; López-López, J.; Monsalve-Guil, L. Implant Treatment by Guided Surgery Supporting Overdentures in Edentulous Mandible Patients. Int J Environ Res Public Health 2021, 18, doi:10.3390/ijerph182211836.
- Velasco-Ortega, E.; Cracel-Lopes, J.L.; Matos-Garrido, N.; Jiménez-Guerra, A.; Ortiz-Garcia, I.; Moreno-Muñoz, J.; Núñez-Márquez, E.; Rondón-Romero, J.L.; López-López, J.; Monsalve-Guil, L. Immediate Functional Loading with Full-Arch Fixed Implant-Retained Rehabilitation in Periodontal Patients: Clinical Study. Int J Environ Res Public Health 2022, 19, doi:10.3390/ijerph192013162.
- Velasco-Ortega, E.; Del Rocío Jiménez-Martin, I.; Moreno-Muñoz, J.; Núñez-Márquez, E.; Rondón-Romero, J.L.; Cabanillas-Balsera, D.; Jiménez-Guerra, Á.; Ortiz-García, I.; López-López, J.; Monsalve-Guil, L. Long-Term Treatment Outcomes of Implant Prostheses in Partially and Totally Edentulous Patients. Materials (Basel) 2022, 15, doi:10.3390/ma15144910.
- D'Haese, J.; Van De Velde, T.; Elaut, L.; De Bruyn, H. A prospective study on the accuracy of mucosally supported stereolithographic surgical guides in fully edentulous maxillae. Clin Implant Dent Relat Res 2012, 14, 293-303, doi:10.1111/j.1708-8208.2009.00255.x.
- Derksen, W.; Wismeijer, D.; Flügge, T.; Hassan, B.; Tahmaseb, A. The accuracy of computer-guided implant surgery with tooth-supported, digitally designed drill guides based on CBCT and intraoral scanning. A prospective cohort study. Clin Oral Implants Res 2019, 30, 1005-1015, doi:10.1111/clr.13514.
- Vogl, S.; Stopper, M.; Hof, M.; Wegscheider, W.A.; Lorenzoni, M. Immediate Occlusal versus Non-Occlusal Loading of Implants: A Randomized Clinical Pilot Study. Clin Implant Dent Relat Res 2015, 17, 589-597, doi:10.1111/cid.12157.
- Lopes, A.; Maló, P.; de Araújo Nobre, M.; Sanchez-Fernández, E. The NobelGuide® All-on-4® Treatment Concept for Rehabilitation of Edentulous Jaws: A Prospective Report on Medium- and Long-Term Outcomes. Clin Implant Dent Relat Res 2015, 17 Suppl 2, e406-416, doi:10.1111/cid.12260.
- Yamada, J.; Kori, H.; Tsukiyama, Y.; Matsushita, Y.; Kamo, M.; Koyano, K. Immediate loading of complete-arch fixed prostheses for edentulous maxillae after flapless guided implant placement: a 1-year prospective clinical study. Int J Oral Maxillofac Implants 2015, 30, 184-193, doi:10.11607/jomi.3679.
- Lopes, A.; Maló, P.; de Araújo Nobre, M.; Sánchez-Fernández, E.; Gravito, I. The NobelGuide(®) All-on-4(®) Treatment Concept for Rehabilitation of Edentulous Jaws: A Retrospective Report on the 7-Years Clinical and 5-Years Radiographic Outcomes. Clin Implant Dent Relat Res 2017, 19, 233-244, doi:10.1111/cid.12456.
- Marra, R.; Acocella, A.; Rispoli, A.; Sacco, R.; Ganz, S.D.; Blasi, A. Full-mouth rehabilitation with immediate loading of implants inserted with computer-guided flap-less surgery: a 3-year multicenter clinical evaluation with oral health impact profile. Implant Dent 2013, 22, 444-452, doi:10.1097/ID.0b013e31829f1f7f.
- Meloni, S.M.; De Riu, G.; Pisano, M.; Cattina, G.; Tullio, A. Implant treatment software planning and guided flapless surgery with immediate provisional prosthesis delivery in the fully edentulous maxilla. A retrospective analysis of 15 consecutively treated patients. Eur J Oral Implantol 2010, 3, 245-251.
- Ciabattoni, G.; Acocella, A.; Sacco, R. Immediately restored full arch-fixed prosthesis on implants placed in both healed and fresh extraction sockets after computer-planned flapless guided surgery. A 3-year follow-up study. Clin Implant Dent Relat Res 2017, 19, 997-1008, doi:10.1111/cid.12550.
- Polizzi, G.; Cantoni, T. Five-year follow-up of immediate fixed restorations of maxillary implants inserted in both fresh extraction and healed sites using the NobelGuide™ system. Clin Implant Dent Relat Res 2015, 17, 221-233, doi:10.1111/cid.12102.
- Pozzi, A.; Sannino, G.; Barlattani, A. Minimally invasive treatment of the atrophic posterior maxilla: a proof-of-concept prospective study with a follow-up of between 36 and 54 months. J Prosthet Dent 2012, 108, 286-297, doi:10.1016/s0022-3913(12)60178-4.

Reviewer 2 Report
Dear Authors,
The topic of the review manuscript titled: Guided dental implant surgery: systematic review is interesting, updated and well written. Before publishing please consider major following concerns:
- Material and Methods: please add section of Risk of Bias assessment
- for discussing section will be beneficial to have more discussion between implant placement in lower and upper jaw as well as in the frontal part (incisors) and molars
- the whole manuscript need English corrections (spaces, capital letter and other, including title and tables)
Author Response
Dear Authors,
The topic of the review manuscript titled: Guided dental implant surgery: systematic review is interesting, updated and well written. Before publishing please consider major following concerns:
- Material and Methods: please add section of Risk of Bias assessment
- for discussing section will be beneficial to have more discussion between implant placement in lower and upper jaw as well as in the frontal part (incisors) and molars
- the whole manuscript need English corrections (spaces, capital letter and other, including title and tables)
Answer
Thank you for reviewing the manuscript, your suggestions have been very helpful in improving the manuscript. all changes and additions requested by you are shown in green
reviewer
- Material and Methods: please add section of Risk of Bias assessment
Answer
risk of bias assessment has been add in materials e methods ed in Results
2.5 Risk of Bias
The ROBINS-I tool was used to calculate the risk of bias and it was evaluated by the 2 reviewers (M.D and A.B.) appointed to select the studies, the studies with a high risk of bias were excluded from the review.
Results
The risk of bias was considered low for all studies included in the review, the following parameters were adopted (ROBINS-I) table 8: Bias due to confounding, Bias in selection of participants into the study, Bias in classification of interventions, Bias due to deviations from intended interventions, Bias due to missing data, Bias in measurement of outcomes, Bias in selection of the reported result: for each parameter the evaluation could be, Low risk , Moderate risk , Serious risk , Critical risk.
Table 8. Risk of bias; Low risk +, Moderate risk -, Serious risk x, Critical risk !
|
|
Bias due to confounding |
Bias in selection of participants into the study |
Bias in classification of interventions |
Bias due to deviations from intended interventions |
Bias due to missing data |
Bias in measurement of outcomes |
Bias in selection of the reported result |
Overall Bias |
|
Ciabattoni et al., 2017 [16]; |
+ |
+ |
+ |
+ |
+ |
+ |
+ |
+ |
|
Derksen et al., 2019 [17] |
+ |
+ |
+ |
+ |
+ |
+ |
+ |
+ |
|
Lopes et al., 2015 [7] |
+ |
+ |
+ |
+ |
+ |
+ |
+ |
+ |
|
Marra et al., 2013 [18] |
+ |
+ |
+ |
+ |
+ |
+ |
+ |
+ |
|
Meloni et al., 2010 [15] |
+ |
+ |
+ |
+ |
? |
+ |
+ |
+ |
|
Polizzi and Cantoni, 2015 [14] |
+ |
+ |
+ |
+ |
+ |
+ |
+ |
+ |
|
Pozzi et al., 2012 [19] |
+ |
+ |
+ |
+ |
+ |
+ |
+ |
+ |
|
Vogl et al., 2015 [20] |
+ |
+ |
+ |
+ |
+ |
+ |
+ |
+ |
|
Yamada et al., 2015 [21] |
+ |
+ |
+ |
+ |
+ |
+ |
+ |
+ |
- for discussing section will be beneficial to have more discussion between implant placement in lower and upper jaw as well as in the frontal part (incisors) and molars
Answer
A further table has been added in the results that identifies the numbers of failures differentiating them from anterior or posterior and mandibular or maxillary location, and the discussion about it has been implemented.
The results regarding the location (mandibular or maxillary, anterior or posterior) on the implant failures are shown below in table 7. Only 4 studies placed implants in both the mandibular and maxillary locations reporting the data on the failures
Table 7. Number of failures in relation to the position of the dental implants, Anterior sinus wall implants (ASW), posterior sinus wall implants (PSW), Right (R), Left (L), data not reported (/).
|
First autor, Data |
Position |
Implant failure |
Anterior Position (number of implant failures) |
Posterior position (number of implant failures) |
Maxilla (number of implant failures) |
Mandibular (number of implant failures) |
|
Ciabattoni et al., 2017 [16]; |
Maxilla\ mandible |
7 |
\ |
\ |
5\193 |
2\90 |
|
Derksen et al., 2019 [17] |
Maxilla\ mandible |
1 |
\ |
\ |
66 |
1\79 |
|
Lopes et al., 2015 [7] |
Maxilla\ mandible |
3 |
Maxilla (lateral incisor) |
2Maxilla (first molar, second premolar) |
3\72 |
0\20 |
|
Marra et al., 2013 [18] |
Maxilla\ mandible |
6 |
\ |
\ |
5\177 |
1\135 |
|
Meloni et al., 2010 [15] |
Maxilla |
2 |
\ |
\ |
2 |
\ |
|
Polizzi and Cantoni, 2015 [14] |
Maxilla |
4 |
\ |
\ |
4 |
\ |
|
Pozzi et al., 2012 [19] |
Maxilla |
3 |
2 ASW |
1 PSW |
3 |
\ |
|
Vogl et al., 2015 [20] |
Mandible |
\ |
\ |
\ |
\ |
\ |
|
Yamada et al., 2015 [21] |
Maxilla |
4 |
2 maxilla (incisive lateral R and incisive lateral L) |
2 maxilla (first molar R and first molar L) |
4 |
\ |
Concerning the position of the implant failures we have only a number of 4 studies that report the data both on the mandibular and maxillary positioning, from these 4 studies the total number of maxillary implants was 508 with a number of failures equal to 13 while the mandibular ones was 324 with 4 failures; it was decided not to perform a meta-analysis of the data for the small number of potentially included studies; The data on the placement of the implants on the anterior and posterior sectors for the included studies are incomplete, and the data on the reported failures would indicate a homogeneous distribution of the failures between the 2 sectors (5 posterior and 4 anterior) further evaluations would be useless due to the lack of reported data in studies
